# Multiple Adenosine-Dopamine (A2A-D2 Like) Heteroreceptor Complexes in the Brain and Their Role in Schizophrenia

**DOI:** 10.3390/cells9051077

**Published:** 2020-04-27

**Authors:** Dasiel O. Borroto-Escuela, Luca Ferraro, Manuel Narvaez, Sergio Tanganelli, Sarah Beggiato, Fang Liu, Alicia Rivera, Kjell Fuxe

**Affiliations:** 1Department of Neuroscience, Karolinska Institutet, 17170 Stockholm, Sweden; 2Observatorio Cubano de Neurociencias, Grupo Bohío-Estudio, 62100 Yaguajay, Cuba; 3Department of Life sciences and Biotechnology, University of Ferrara, 44121 Ferrara, Italy; frl@unife.it (L.F.); bggsrh@unife.it (S.B.); 4Facultad de Medicina, Instituto de Investigacion de Málaga, Universidad de Malaga, 29016 Malaga, Spain; mnarvaez@uma.es; 5Department of Medical Sciences, University of Ferrara, 44121 Ferrara, Italy; tgs@unife.it; 6Campbell Research, Centre for Addiction and Mental Health Institute, University of Toronto, Toronto, ON MS5 1A1, Canada; Fang.Liu@camh.ca; 7Department of Cell Biology, University of Malaga, Instituto de Investigación Biomédica (IBIMA), 29016 Malaga, Spain; arivera@uma.es

**Keywords:** adenosine receptors, A_2A_-D_2_ heteroreceptor complexes, schizophrenia, brain, novel pharmacology, heterobivalent drugs, sigma 1 receptor

## Abstract

In the 1980s and 1990s, the concept was introduced that molecular integration in the Central Nervous System could develop through allosteric receptor–receptor interactions in heteroreceptor complexes presents in neurons. A number of adenosine–dopamine heteroreceptor complexes were identified that lead to the A_2A_-D_2_ heteromer hypothesis of schizophrenia. The hypothesis is based on strong antagonistic A_2A_-D_2_ receptor–receptor interactions and their presence in the ventral striato-pallidal GABA anti-reward neurons leading to reduction of positive symptoms. Other types of adenosine A_2A_ heteroreceptor complexes are also discussed in relation to this disease, such as A_2A_-D_3_ and A_2A_-D_4_ heteroreceptor complexes as well as higher order A_2A_-D_2_-mGluR5 and A_2A_-D_2_-Sigma1R heteroreceptor complexes. The A_2A_ receptor protomer can likely modulate the function of the D_4_ receptors of relevance for understanding cognitive dysfunction in schizophrenia. A_2A_-D_2_-mGluR5 complex is of interest since upon A_2A_/mGluR5 coactivation they appear to synergize in producing strong inhibition of the D2 receptor protomer. For understanding the future of the schizophrenia treatment, the vulnerability of the current A_2A_-D_2_like receptor complexes will be tested in animal models of schizophrenia. A_2A_-D_2_-Simag1R complexes hold the highest promise through Sigma1R enhancement of inhibition of D2R function. In line with this work, Lara proposed a highly relevant role of adenosine for neurobiology of schizophrenia.

## 1. Introduction

The concept was developed in the 1990s that the communication in the central dopamine neurons could become integrated with adenosine communication through the formation of heteroreceptor complexes built up of adenosine and dopamine receptors in a receptor subtype specific way located in the plasma membrane, specially extrasynaptic regions [1]. There exist two main receptors for the modulator of adenosine in the brain, namely adenosine A_1_ and A_2A_ receptors. The major dopamine receptors in the brain are the D_1_ and D_2_ receptors but also D_3_ and D_4_ receptors play an important role in the brain [2]. Over the years a number of adenosine–dopmaine heteroreceptor complexes have been found, inter alia A_2A_-D_2_, A_1_-D_1,_ A_2A_-D_3_, and A_2A_-D_4_ heteroreceptor complexes besides the dopamine and adenosine homoreceptor complexes, see [2,3]. They illustrated the diversity and specificity of the former heteroreceptor complexes and their impact on molecular integration in the brain. 

Adenosine represents an endogenous modulator of the neuronal and astroglia networks of the Central Nervous System (CNS) [2,4] acting via volume transmission [5,6]. Its concentration is dependent on the synthesis and breakdown of ATP which is metabolized to adenosine monophosphate (AMP) by adenosine kinase. The action of 5′-nucleotididase then removes the monophosphate from AMP and adenosine is formed. The intracellular and extracellular concentrations of adenosine are in equilibrium with each other through transporters. The inhibitory A_1_ receptors have a widespread distribution in the brain and are especially enriched in the neocortex, the hippocampus, and cerellum [7,8]. Additionally in the striatum, they are mainly located in the striato-entopeduncular/nigral GABA neurons, called the direct pathway [1]. In contrast, the A_2A_ receptors are found in highest densities in the dorsal and ventral striato-pallidal GABA neurons [2,3,9,10]. 

The pharmacological treatment of schizophrenia did not begin with targeting the adenosine receptor but with targeting the dopamine (DA) receptors and the major receptor turned out to be the D_2_ receptor [11,12,13,14,15,16]. The DA receptor in the meso-limbic system was proposed to be a major target [17]. This research represented the introduction of the DA hypothesis of schizophrenia. The increased activity in the meso-limbic DA neurons was explained on the basis of the combined glutamate/DA hypothesis of schizophrenia [17,18]. A reduced NMDA receptor function in the descending glutamate cortical systems to the ventral tegmental area (VTA) was proposed to lead to a reduced activation of their GABA interneurons associated with increased activation of the meso-accumbal DA neurons. As a result, the D_2_ receptor induced inhibition of the ventral striato-pallidal anti-reward GABA neurons which were enhanced with a reduced glutamate drive reaching the frontal cortex from the mediodorsal glutamate neurons [19]. The mixed hypothesis of DA/glutamate interactions was also beautifully advanced by Fang Liu and colleagues who demonstrated that NMDA receptors and D_2_ receptor can form heteroreceptor complexes in glutamate synapses in which the D_2_ receptor protomer, upon activation, can significantly inhibit NMDA receptor signaling [20].

The clinical work on schizophrenia gives evidence that it is mainly the positive symptoms that are blocked after treatment with D_2_ receptor antagonists, like hallucinations and delusions. Instead, cognitive and negative symptoms with lack of social interactions are more difficult to counteract.

The next step regarding the development of novel strategies for treatment of schizophrenia was the demonstration that A_2A_ receptor agonists can ex vivo strongly reduce the affinity of the D_2_ receptor agonist binding sites in the high affinity state through antagonistic allosteric receptor–receptor interactions [19,21,22]. Of importance was also the demonstration using FRET and BRET, coimmunoprecipitation and in situ Proximity Ligation Assay (in situ PLA) that A_2A_-D_2_ heteroreceptor complexes can be formed [23,24,25,26,27]. It was also demonstrated that coaggregation and cointernalization can develop [28,29]. 

It was of high interest that the A_2A_ receptor agonist could be established to be an atypical antipsychotic drug in phencyclidine and amphetamine models of schizophrenia [30]. These results lead to the hypothesis that A_2A_ receptor agonists can be novel antipsychotic drugs by activating the antagonistic allosteric receptor–receptor interactions in the A_2A_-D_2_ heteroreceptor complexes located mainly in the ventral striato-pallidal GABA pathway. It represents an anti-reward pathway which is overactivated in schizophrenia due to increased activation of its D_2_ receptors. The inhibition of the D_2_ receptor function by A_2A_ receptor activation in this receptor complex can restore the glutamate drive to the frontal cortex from the medial dorsal thalamic nucleus [19,31].

In the current review, the A_2A_-D_2_ like heteroreceptor complexes will be discussed in detail in relation to schizophrenia. In addition, the potential role in schizophrenia of the adenosine A_2A_ isoreceptor complexes, specially the A_1_-A_2A_ and A_2A_-A_2B_ complexes, will be covered [32,33,34]. Additionally, other types of adenosine A_2A_R heteroreceptor complexes will be covered in relation to this disease such as A_2A_-D_3_ receptor [35] and A_2A_-D_4_ receptor complexes [36] as well as higher order A_2A_R-D_2_R-mGluR5 [37], and A_2A_R-D_2_R-Sigma1R [22,38] heteroreceptor complexes (see Figure 1). Can other types of A_2A_ receptor complexes be involved in schizophrenia? Other adenosine hypotheses will also be explored besides the first one on antagonistic A_2A_-D_2_ interactions in heteroreceptor complexes in the nucleus accumbens bringing down activity in the overactive D_2_ receptor protomer in schizophrenia [2,19,31,39,40].

## 2. The A_2A_-D_2_ Receptor Heteromer Hypothesis and the LARA et al. Adenosine Hypothesis of Schizophrenia

The A_2A_-D_2_ heteromer hypothesis of schizophrenia has been introduced and further advanced in a substantial number of papers from our group since the 1990s (see e.g., [1,2,19,31,41]). The most relevant location is its presence in the ventral striato-pallidal GABA neurons of the nucleus accumbens, representing anti-reward neurons, and their glutamate synapses originating mainly from the cerebral cortex. The accumbal pathway regulates the brain circuit to the prefrontal cortex from the ventral pallidum, via the medial dorsal thalamic glutamate system. There is a subnucleus specific loss of these glutamate neurons to the frontal cortex in schizophrenia [42]. It is of interest that D_2_ like receptor agonists microinjected into the nucleus accumbens significantly reduces extracellular glutamate levels in the prefrontal cortex as determined with dual probe microdialysis [1,43,44]. The glutamate drive from the medial dorsal thalamus to the prefrontal cortex is significantly reduced. This is in line with the evidence that D_2_ receptor agonists can enhance psychosis [14]. Of high interest is the evidence that microinjections of quinpirole (D_2_ receptor like agonist) together with the A_2A_ receptor agonist CGS21680 into the nucleus accumbens block the reduction of extracellular glutamate levels in the prefrontal cortex by quinpirole and even lead to a significant A_2A_ receptor agonist induced increase of extracellular glutamate levels in this region [19]. These results were strengthened by the demonstration that the A_2A_ receptor agonist microinjected into the nucleus accumbens counteracted the reduction of extracellular GABA levels induced by the D_2-_like receptor agonist coinjected with the A_2A-_like receptor agonist. In a similar way, the D_2-_like agonist induced increase in the extracellular GABA levels in the medial dorsal thalamic nucleus was counteracted by the A_2A_ receptor agonist [19,45].

There also exists a rat model of schizophrenia based on the amphetamine-induced sensitized state [46]. After an acute amphetamine challenge in the amphetamine sensitized state it was therefore tested if changes developed in the antagonistic A_2A_-D_2_ receptor–receptor interactions in the D_2_ receptor binding affinity of the ventral striatum [47]. Compared with the saline sensitized state, the A_2A_ receptor agonist CGS21680 induced a reinstatement of the antagonistic A_2A_-D_2_ receptor interaction in the ventral striatum but not in the dorsal striatum. These results can help explain the atypical antipsychotic profile of the A_2A_ receptor agonist since the motor effects are produced through actions in the dorsal striatum. In line with this view, the D_2_ receptor homomers dominate in the dorsal striatum in this model of schizophrenia [46]. The major action of amphetamine-induced dopamine release in the dorsal striatum appears to be an enhanced affinity of the D_2_ receptors for each other, leading to the increased formation of D_2_ receptor homomers.

Taken together, the results suggest that the antagonistic allosteric receptor–receptor interactions in A_2A_R-D_2_ heteroreceptor complexes in the ventral striato-pallidal GABA neurons are the major target for the atypical antipsychotic actions of the A_2A_ receptor agonist CGS21680. It leads to a marked reduction of D_2_ receptor protomer recognition and signaling. The dual microdialysis findings provide strong indications that the brake on D_2_ receptor protomer signaling induced by the A_2A_ receptor agonist leads at the brain circuit level to increased activity in the medial dorsal thalamic glutamate system with increased glutamate drive reaching the prefrontal cortex [19,31]. In support of our A_2A_R-D_2_ heteromer hypothesis of schizophrenia [2,19,31,41], an increase in A_2A_ receptor agonist binding sites was found in the striatum of postmortem brains of chronic schizophrenics [48]. Furthermore, the salience dysregulation found in schizophrenia is likely produced by pathological increases in the activity of the meso-limbic DA neurons [49]. This overactivity can be blocked by restoring the brake on the D_2_R protomer signaling by agonist activation of the A_2A_ receptor protomer signaling in several types of A_2A_-D_2_ heteroreceptor complexes on the ventral striato-pallidal GABA anti-reward neurons. In this way, the salience regulation can become normalized.

Another type of purinergic hypothesis of schizophrenia was introduced by Lara and colleagues in 2000 and is of high interest [50,51,52,53]. This model aimed to bring together many aspects of schizophrenia including the role of purines in the immune system. Allopurinol, which reduces purine degradation and increases brain levels of adenosine and inosine, was tested in poorly responsive patients with schizophrenia [51]. The two patients with schizophrenia, tested with allopurinol, were clinically improved by this treatment which gives clinical support for their hypothesis. The same year they made the interesting observation that chronic treatment with clozapine, but not with haloperidol increased striatal ectonucleotidase activity. Based on their findings this likely leads to increases in adenosine levels and activation of inter alia adenosine A_2A_ receptors [52], which can bring down the function of overactive D_2_ receptors through antagonistic receptor–receptor interactions. In their 2006 paper, Lara and colleagues proposed the highly relevant role of adenosine for the neurobiology of schizophrenia and its treatment [50]. Pharmacological treatment to increase the formation of adenosine in the brain is proposed by Lara and colleagues to be a relevant strategy using, for example, allopurinol which is well tolerated in patients with schizophrenia. In the future it will be of high interest to compare the effects of, for example, allopurinol and increased ectonucleotidase activity with effects of adenosine A_1_ receptor and/or A_2A_ receptor agonists in the treatment of schizophrenia.

In 2011, one more adenosine hypothesis of schizophrenia was introduced [54]. Multiple adenosine targets were emphasized which involved also adenosine A_1_ and A_2A_ receptor agonists and the enzyme adenosine kinase.

## 3. A_1_-A_2A_ Isoreceptor Complexes

A major finding was the location of this receptor complex in the striatal glutamate nerve terminals [32]. At high concentrations adenosine activated the A_2A_R protomers, which through antagonistic A_2A_-A_1_ receptor–receptor interaction could reduce the inhibitory action of A_1_ receptor protomer on glutamate release through reduction of its affinity for A_1_ receptor agonists. Instead the facilitatory actions of the A_2A_ receptor became dominant and enhancement of glutamate release was observed [32,34]. The available information indicates that the A_2A_ receptor protomer has their major role in reducing symptoms of schizophrenia mainly through enhancing activity of the ventral anti-reward striato-pallidal GABA pathway (see above). It therefore seems possible that such actions can also develop through the activation of the A_2A_ receptor protomer in the prejunctional A_1_-A_2A_ receptor complex as described in this paragraph. It seems to develop through enhanced release of glutamate, enhancing the effects of the postjunctional antagonistic A_2A_-D_2_ receptor interaction also leading to increased activity GABA anti-reward neurons.

## 4. A_2A_-A_2B_ Isorecepor Complexes

These receptor complexes were demonstrated with a number of techniques such as FRET, BRET, bimolecular complementation, and in situ Proximity Ligation Assay (in situ PLA) [33,55]. The most exciting finding was the observations that the A_2B_ receptor protomer blocked the A_2A_ receptor protomer ligand recognition and signaling [33]. The pharmacology of the A_2A_ receptor was markedly lost since there was no high affinity for A_2A_R ligands at the A_2A_ receptor protomer. The potency of A_2A_ receptor ligands to increase cAMP levels was also dramatically reduced.

These results also lead to new understanding for how schizophrenia can develop based on an excessive formation of A_2A_-A_2B_ heteroreceptor complexes reducing the formation of the A_2A_-D_2_ heteroreceptor complexes in the ventral striato-pallidal GABA neurons. We assume the existence of a specific balance between different types of A_2A_ isoreceptor and/or A_2A_ heteroreceptor complexes in cortical and striatal subregions of the brain, involving both neuronal and/or glial cells [31,55,56]. Changes in the balance may in part have a genetic origin due to changes in the expression pattern of the A_2A_ vs. the A_2B_ and the D_2_ receptor, especially in the ventral striato-pallidal GABA anti-reward neurons [55,56]. Turning off the activity of the A_2A_ receptor by increased formation of A_2A_-A_2B_ receptor complexes should certainly increase the D_2_ receptor signaling with increased affinity for DA in the high affinity state [3,57], especially if higher order A_2A_-A_2B_-D_2_ receptor complexes may exist. The development of D_2_ receptor supersensitivity [58] in this way should result in a block of anti-reward information of the ventral striato-pallidal GABA neurons from reaching the frontal cortex due to inhibition of the medial dorsal thalamic nucleus [22]. As a result, due to the lack of this filter mechanism, all anti-reward stimuli from this brain circuit are blocked from reaching the frontal cortex. Such a failure can contribute to development of psychotic symptoms in schizophrenia, since even irrelevant stimuli become regarded as salient.

## 5. A_2A_-D_3_ Heteroreceptor Complexes

The first evidence for their existence was obtained using FRET analysis [35] with a significant FRET efficiency demonstrated in the A_2A_R-D_3_R complex in the plasma membrane. Like in the A_2A_-D_2_ receptor complex, positively charged arginin epitopes were found in the third intracellular loop of the D_3_R interacting with negatively charged epitopes in the A_2A_ located in its C-terminus [2,25]. Again, like in the case of the A_2A_R-D_2_, the A_2A_ receptor agonist reduced the affinity of the high affinity binding site of the D_3_ receptor. The A_2A_ receptor agonist CGS21680 also blocked the DA induced inhibition of the cAMP accumulation produced by activation of the D_3_ receptor [35]. Thus, it appears clear that there exists an allosteric antagonistic receptor–receptor interaction also in the A_2A_-D_3_ heteroreceptor complex. The D_3_ receptors are shown to exist in the brain especially in the frontal cortex, the nucleus accumbens, and the ventral midbrain, and are involved in motivation, emotions, and reward and anti-reward, but the A_2A_-D_3_ receptor complexes have not yet been analized in the brain.

Through the careful and excellent work of Pierre Sokoloff and Bernard Le Foll [59], we know that D_3_ receptor selective compounds are available for clinical trials in schizophrenia. By use of PET, these D_3_ selective compounds were shown to occupy D_3_ receptors in vivo. As to mechanisms involved in the D_3_R modulation of brain function, it can involve, for example, modulation of the glutamate projections from the prefrontal cortex to the nucleus accumbens and of dopamine nerve cell-dendritic networks in the ventral tegmental area rich in DA nerve cells projecting inter alia into the nucleus accumbens, the prefrontal cortex and the cingulate cortex. It will therefore be of substantial interest to demonstrate and understand the role of the A_2A_-D_3_ receptor complexes in these brain circuits.

## 6. A_2A_-D_4_ Heteroreceptor Complexes

It is of high interest that very recently the formation was demonstrated of A_2A_-D_4_ heteroreceptor complexes in different regions of the rat forebrain using in situ PLA [36]. Thus, A_2A_ receptors can likely modulate the function also of the D_4_ receptors. It should be underlined that D_4_ receptor specific antagonists have not able to reduce antipsychotic effects in treatment of schizophrenia [60,61]. Instead the D_4_ receptors have a role in cognitive function and in modulating gamma oscillations [62]. In fact, the D_4_ receptor has a potential for reducing the cognitive deficits in schizophrenia. There are also positive interactions found in the ability of D_4_ receptors to interact with neuroregulin/ErbB4 in the modulation of the activity of GABA parvalbumin positive interneurons in the prefrontal cortex and the hippocampus that regulate gamma oscillations and cognitive function [62]. Thus, there is the possibility that D_4_ agonists/antagonists, which alter D_4_ receptor function, can improve cognition deficits in schizophrenia. It will be highly interesting to see how A_2A_ receptor protomers in the A_2A_-D_4_ complexes can modulate the cognition effects of the D_4_ receptor protomer in schizophrenia, especially in regions with high densities of the A_2A_-D_4_ receptor complexes.

## 7. A2AR-D2R-mGlu5R Heteroreceptor Complexes

Several groups have been involved in finding and working on these heteroreceptor complexes located especially in the soma-dendritic regions of the ventral striato-pallidal anti-reward GABA neurons [19,31,37,40,63,64]. The early work in 2001 [65] indicated the existence of A_2A_-mGluR5 heteroreceptor complexes which demonstrated enhanced inhibition of D_2_ receptor function in the GABA anti-reward neurons. In 2009, evidence was obtained in cellular models that higher order A_2A_-D_2_-mGluR5 heteroreceptor complexes exist based on BRET and bimolecular fluorescence complementation [37]. The trimeric heteroreceptor complex appeared to be located mainly in perisynaptic locations in the ventral cortico-striatal glutamate synapses. The A_2A_ and mGluR5 receptor protomer upon coactivation appeared to synergize to produce enhanced inhibition of the D_2_R protomer recognition and signaling [37]. Such a strong inhibition of D_2_ signaling [66] may be necessary when the D_2_ shows a high increase in its activity in schizophrenia. Heterobivalent drugs can here be useful in the treatment of schizophrenia by using one A_2A_ receptor agonist pharmacophore in combination with a mGluR5 positive allosteric modulator pharmacophore to specifically target this receptor complex and inhibit the overactive D_2_ receptor protomer signaling. 

## 8. A_2A_-D_2_-Sigma1 Heteroreceptor Complexes

This trimeric complex is proposed to exist based on many findings, including the existence of A_2A_-D_2_-Sigma1 heteroreceptor complexes [38,40,67,68,69,70], and are of high interest. Thus, the participation of the chaperone protein Sigma1R in this receptor complex leads to a strong increase in the ability of the A_2A_ receptor protomer to inhibit the recognition and signaling of the D_2_ receptor protomer [22]. The cocaine-induced increase in the density of the Sigma1R plays a relevant role as well as its ability to recruit the Sigma1R to the plasma membrane [71]. It remains to be demonstrated if the inhibition of the D_2_ receptor protomer function by the A_2A_R becomes too strong in the presence of Sigma1R to be used in treatment of schizophrenia (see Figure 2). However, it may be that in the absence of cocaine, the enhanced allosteric A_2A_R mediated inhibition of the D_2_R protomer function produced by the Sigma1R may be sufficient without leading to a pathological silence of the D_2_R function (see Figure 2, panel B). If this is true, this research can lead to a new strategy for treatment of schizophrenia.

## 9. Conclusions and Future Work

As a new strategy of understanding schizophrenia and its treatment, the vulnerability of the current A_2A_-D_2_like receptor complexes will be tested in models of schizophrenia in terms of changes in their density, composition, and stoichiometry as well as in their allosteric receptor–receptor interactions. The current results open up the possibility that the neurochemical results may be linked to schizophrenia-like behavior. In future work it will be of special interest to see which of the changes found in the various A_2A_-D_2_like heteroreceptor complexes can correlate to the positive and/or negative symptoms found in models of schizophrenia. The A_2A_-D_2_-Sigma1 receptor complex appears to hold the highest promise. This is based on our hypothesis that A_2A_ receptor agonists may strengthen their inhibitory effects on overactive D_2_ functions in the GABA anti-reward neurons in the presence of sigma1R in the heteroreceptor complex. Thus, A_2A_ receptor agonist treatment with or without co-treatment with Sigma1 receptor agonists should strongly enhance their antagonistic allosteric A_2A_-D_2_ receptor interactions. As a consequence, the positive symptoms of schizophrenia should be significantly reduced.

## Figures and Tables

**Figure 1 cells-09-01077-f001:**
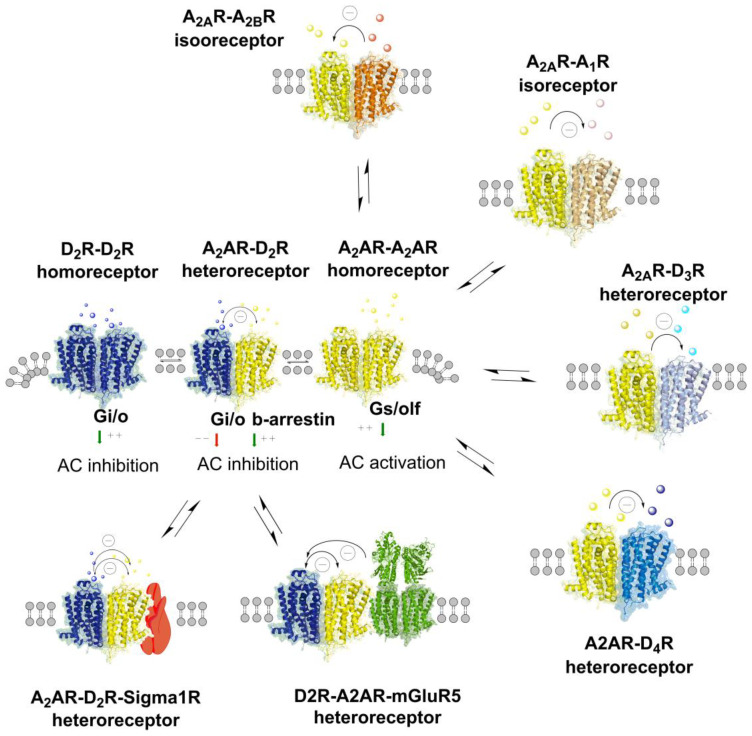
A_2A_-D_2_ heteroreceptor complexes, A_2A_ isoreceptor complexes, and higher order A_2A_-D_2_ heteroreceptor complexes are illustrated and exist mainly in the ventral and dorsal striatum. Their balance with the A_2A_ and D_2_ homoreceptotor complexes are indicated as well as their allosteric receptor–receptor interactions. The nature of the allosteric receptor–receptor interactions in each complex is provided in the top part of the receptor complex. Antagonistic allosteric modulation is indicated as (-) and facilitatory allosteric modulation as (+).

**Figure 2 cells-09-01077-f002:**
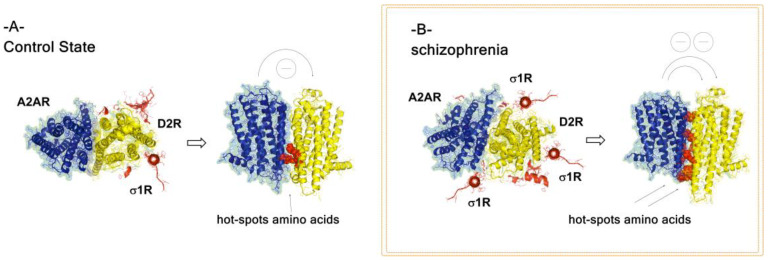
Proposed alterations of A_2A_-D_2_-Sigma1 heteroreceptor complexes in nucleus accumbens in schizophrenia. (**A**) A_2A_-D_2_-Sigma1 higher-order heteroreceptor complexes given in a control state. The adaptor protein Sigma1receptor is given in red. (**B**) In schizophrenia there may develop an increased drive in the accumbal D_2_ receptor protomer signaling as discussed. This hyperactivity may be counteracted by Sigma1R activation (e.g., increased density and/or treatment with Sigma1R agonists) due to its ability to enhance the allosteric A_2A_ receptor inhibition of the D_2_ receptor protomer signaling and recognition in this heterotrimeric receptor complex. This increased inhibition may be brought about through an increased number of hotspot amino acids formed in the A_2A_-D_2_ receptor interface, outlined as red filled spaces.

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
