# Peer review of "Multiple Adenosine-Dopamine (A2A-D2 Like) Heteroreceptor Complexes in the Brain and Their Role in Schizophrenia"

_cells, 2020, doi:10.3390/cells9051077_

Round 1

Reviewer 1 Report

The paper presents an interesting review of receptor-receptor interactions in A2A-D2like heteroreceptor complexes in the ventral striato-pallidal GABA anti-reward neurons and the possible usage of A2AR agonist CGS21680 in the treatment of positive, negative and cognitive symptoms of schizophrenia. The paper is very well written, and the authors included the most important findings in the topic.

Author Response

We appreciate the excellent comments of reviewer 1. We have added and improved the manuscript in the Introductory section.

Reviewer 2 Report

  1. I’ve some issues with abstract and introduction. In a significant part it represents a composition of sentences selected from various publications. Single sentences make sense, but their composition into a coherent whole is in my opinion is not too bright. I suppose, issue is lack of a clear concept of narration – starting from the whole idea to nitty-gritty elements of the manuscript. Such an approach results in a huge mess, both as regards conceptual and compositional aspects of the abstract and introduction. For introduction, I suggest starting from short characterization of adenosine and dopamine, and their receptors. Next, some of aspects of homo and heteroreceptors should be presented. Finally, introduction may go to hypothesis of schizophrenia.
  2. I find some trouble with references No 42-67, they are in the text but not be cite in the reference list. The list of reference should be arranged in better way.

Remarks:

  1. Abstract, line 26: “Another type of purinergic hypothesis of schizophrenia was introduced by Lara and colleagues and is of high interest”.

Comment: is not clear, beside the point

  1. Line 46-47 “The inhibitory A1 receptors have a widespread distribution in the brain and are especially enriched in the neocortex, the hippocampus and cerellum”.

Comment: too general and not linked to the previous sentence

  1. Line 139-140: “In support of our hypothesis an increase in A2A receptor agonist binding sites was found in the striatum of postmortem brains of chronic schizophrenics [44]”.

Comment: who are the authors? OUR?

Author Response

Answer to reviewer

1. I’ve some issues with abstract and introduction. In a significant part it represents a composition of sentences selected from various publications. Single sentences make sense, but their composition into a coherent whole is in my opinion is not too bright. I suppose, issue is lack of a clear concept of narration – starting from the whole idea to nitty-gritty elements of the manuscript. Such an approach results in a huge mess, both as regards conceptual and compositional aspects of the abstract and introduction. For introduction, I suggest starting from short characterization of adenosine and dopamine, and their receptors. Next, some of aspects of homo and heteroreceptors should be presented. Finally, introduction may go to hypothesis of schizophrenia.

We appreciate the excellent comments of reviewer 1. We have followed your suggestions and improved the manuscript accordingly. The Abstract and Introduction were rewritten.

2. I find some trouble with references No 42-67, they are in the text but not be cite in the reference list. The list of reference should be arranged in better way.

Thanks. The reference list is now correct.

3. Abstract, line 26: “Another type of purinergic hypothesis of schizophrenia was introduced by Lara and colleagues and is of high interest”.

Comment: is not clear, beside the point

Thanks for this suggestion. The abstract was rewritten.

4. Line 46-47 “The inhibitory A1 receptors have a widespread distribution in the brain and are especially enriched in the neocortex, the hippocampus and cerellum”.

Comment: too general and not linked to the previous sentence

Thanks. The sentence was rewritten and proper references were added.

5. Line 139-140: “In support of our hypothesis an increase in A2A receptor agonist binding sites was found in the striatum of postmortem brains of chronic schizophrenics [44]”.

Comment: who are the authors? OUR?

Thanks. The sentence was rewritten